# RNA-Binding Proteins as Important Regulators of Long Non-Coding RNAs in Cancer

**DOI:** 10.3390/ijms21082969

**Published:** 2020-04-23

**Authors:** Katharina Jonas, George A. Calin, Martin Pichler

**Affiliations:** 1Division of Oncology, Department of Internal Medicine, Medical University of Graz (MUG), 8036 Graz, Austria; katharina.jonas@medunigraz.at; 2Research Unit for Non-Coding RNAs and Genome Editing, Medical University of Graz (MUG), 8036 Graz, Austria; 3Department of Experimental Therapeutics, The University of Texas MD Anderson Cancer Center, Houston, TX 77030, USA; gcalin@mdanderson.org

**Keywords:** long non-coding RNAs (lncRNAs), RNA-binding proteins (RBPs), cancer, post-transcriptional regulation, RNA stability

## Abstract

The majority of the genome is transcribed into pieces of non-(protein) coding RNA, among which long non-coding RNAs (lncRNAs) constitute a large group of particularly versatile molecules that govern basic cellular processes including transcription, splicing, RNA stability, and translation. The frequent deregulation of numerous lncRNAs in cancer is known to contribute to virtually all hallmarks of cancer. An important regulatory mechanism of lncRNAs is the post-transcriptional regulation mediated by RNA-binding proteins (RBPs). So far, however, only a small number of known cancer-associated lncRNAs have been found to be regulated by the interaction with RBPs like human antigen R (HuR), ARE/poly(U)-binding/degradation factor 1 (AUF1), insulin-like growth factor 2 mRNA-binding protein 1 (IGF2BP1), and tristetraprolin (TTP). These RBPs regulate, by various means, two aspects in particular, namely the stability and the localization of lncRNAs. Importantly, these RBPs themselves are commonly deregulated in cancer and might thus play a major role in the deregulation of cancer-related lncRNAs. There are, however, still many open questions, for example regarding the context specificity of these regulatory mechanisms that, in part, is based on the synergistic or competitive interaction between different RBPs. There is also a lack of knowledge on how RBPs facilitate the transport of lncRNAs between different cellular compartments.

## 1. Introduction

The concept that RNA merely serves as an intermediate, conveying the genetic information encoded in the form of DNA to be translated into proteins, has long been overthrown [1,2]. In fact, the vast majority of the genome does not code for proteins but is transcribed into various types of so-called non-coding RNAs (ncRNAs), which by themselves fulfill a multitude of pivotal regulatory functions [2,3,4,5]. Among these ncRNAs, one large group with particularly versatile functions are the long non-coding RNAs (lncRNAs), a class of ncRNAs that are defined as being longer than 200 nucleotides [6]. These lncRNAs can originate from different genomic locations. Most are interspersed between protein-coding genes (long intergenic non-coding RNAs, lincRNAs), while others are transcribed from the sense or antisense strands of introns and also exons of coding genes, and yet another type of lncRNAs originates from enhancer regions (eRNA) [7,8,9]. Rather recently it has been discovered that the class of lncRNAs is even more diverse, as it not only includes linear transcripts but also circular RNAs (circRNAs), which are a product of an mRNA splicing process referred to as backsplicing [10].

Compared to protein-coding genes, lncRNAs are poorly conserved between different species and their expression levels are rather low [7,8]. Initially, this led to the belief that they were nothing but transcriptional noise [6,7,8]. Soon, however, it was discovered that lncRNAs do exhibit considerable functionality, for example as regulators of transcription [6,7,8]. Mechanisms of transcriptional regulation by lncRNAs are multifarious and can occur either *in cis* or *in trans*, meaning either closer to or further away from the lncRNA’s site of transcription, respectively [6,7]. A frequent mechanism of transcriptional regulation via lncRNAs is the recruitment, or prevention of such, of components of chromatin or histone-modifying complexes, like polycomb repressive complexes or histone deacetylase complexes [11,12,13]. LncRNAs can also direct transcription factors or cofactors to promoter regions of genes and facilitate the formation of chromatin loops between distant enhancers and promoters, as observed for some eRNAs [9,14,15,16,17]. Another way that they can impact transcription is by interfering with the RNA polymerase II transcription machinery, thereby blocking transcriptional initiation or elongation [18]. LncRNAs are important regulators not only at the level of transcription but also at the post-transcriptional level [6,7]. They regulate pre-mRNA splicing by interacting with splicing factors or with the mRNA itself [19,20,21,22]. A well-studied example for this is metastasis-associated lung adenocarcinoma transcript 1 (MALAT1), an lncRNA that interacts with serine/arginine (SR) splicing factors, thereby influencing their subnuclear distribution, phosphorylation status, and consequently their activity [20]. Apart from splicing, lncRNAs also regulate mRNA stability and translation by interacting with translation factors and proteins involved in mRNA decay, or by acting as so-called competitive endogenous RNAs (ceRNAs) [23,24,25,26,27]. These ceRNAs sponge up microRNAs (miRNAs) and thus prevent them from binding to and inducing degradation or translational repression of their mRNA targets [25,26,27].

According to the NONCODE database (version v5.0), 96,308 lncRNA genes have been identified in the human genome [28]. Steadily increasing focus is now being put on the functional annotation of these lncRNAs, especially regarding their role in disease [28]. Their implications in cancer are of particular interest, as many lncRNAs have been found to be deregulated in cancer [29]. According to lnc2Cancer, a database for human lncRNAs with experimentally supported cancer associations, 1614 different lncRNAs were found to be implicated in a total number of 165 cancer subtypes [30], influencing different aspects of cancer biology like proliferation [31,32,33,34,35], apoptosis [34,35,36], invasion [32,35,36,37], metastasis [38,39,40,41], angiogenesis [41,42,43], drug resistance [35,44], and genome stability [45]. According to lnc2Cancer, the most studied cancer-associated lncRNAs are MALAT1 [46], H19 [47], HOX antisense intergenic RNA (HOTAIR) [48], maternally expressed 3 (MEG3) [49], taurine upregulated 1 (TUG1) [50], antisense non-coding RNA in the INK4 locus (ANRIL) [51] and nuclear-enriched abundant transcript 1 (NEAT1) [52].

While lncRNAs are distinguished from protein-coding mRNAs in several aspects, they also share certain similarities [6,7]. Many of them are, for example, spliced, 3′ polyadenylated, and 5′ capped, just like mRNAs [6,7]. This post-transcriptional regulation is a crucial aspect in the life of both coding and non-coding RNAs and is primarily facilitated by RNA-binding proteins (RBPs), as they dynamically coordinate the maturation, transport, and stability of all types of RNA [53,54]. Thus, changes in the expression level or functionality of RBPs can have profound and far-reaching consequences [53]. In fact, just like lncRNAs, there are numerous RBPs that are altered in cancer [53]. To fully elucidate the role and implications of lncRNAs in cancer it has to be understood how they themselves are regulated. Therefore, this review will put a focus on the regulation of lncRNAs by RBPs in the context of cancer. The RBPs that are discussed in this review (for a summary see Table 1) were thus chosen because they are, just like the lncRNAs they regulate, known to play a role in cancer. Even though these examples do not cover all regulatory interactions between RBPs and lncRNAs they include well-known RBPs and extensively studied cancer-associated lncRNAs and give a comprehensive overview of the two main aspects of post-transcriptional regulation of lncRNAs, namely lncRNA stability on the one hand and lncRNA transport and localization on the other hand.

## 2. RBPs Regulating lncRNA Stability

### 2.1. Human Antigen R (HuR)

The human antigen R (HuR), the protein product of the *ELAV1* gene, is a ubiquitously expressed RBP that contains three RNA recognition motifs (RRMs) via which it preferentially binds to adenylate/uridylate-rich RNA elements (AREs) [71,72,73]. AREs are signals for rapid RNA degradation, and by blocking these recognition sites HuR can stabilize its RNA interaction partners [71,72,73]. HuR is frequently upregulated in cancer cells and is known to be involved in many hallmarks of cancer, such as invasion, angiogenesis, and inflammation, by post-transcriptionally regulating various cancer-related mRNAs [71,72,74,75,76]. HuR not only binds to protein-coding mRNA but also interacts with lncRNAs, thereby influencing their stability both in a positive and negative manner [55,56,57,58,77].

An example of an lncRNA that is well-known to be associated with cancer and whose stability has been found to be enhanced by HuR in ovarian cancer is NEAT1 [55]. NEAT1 exists in two isoforms, NEAT1_1 and NEAT1_2, with lengths of 3.7 kb and 22.7 kb, respectively [25,78]. The longer of the two isoforms, NEAT1_2, is a crucial architectural component of nuclear paraspeckles, which are large ribonucleoprotein (RNP) complexes [25,78]. NEAT1 is, just like HuR, frequently upregulated in many types of cancer, where it contributes to the progression of the disease by acting as a ceRNA sponging up different miRNAs [25,79,80,81]. A study by Chai et al. found NEAT1 also to be upregulated in human ovarian cancer tissue and ovarian cancer cell lines compared to non-cancerous tissue and cell lines [55]. Additionally, they observed similarly elevated mRNA levels of HuR and, based on RNA immunoprecipitation (RIP) results, the authors postulated a direct interaction between NEAT1 and HuR [55]. RIP assays present a simple and widely used method to study RNA–protein interactions (for a comprehensive review on this and other methods to study RNA–protein interactions see [82]) and are based on the use of antibodies to precipitate and isolate a protein of interest together with its associated RNAs [82]. Furthermore, they showed that overexpression of HuR in an ovarian carcinoma cell line (OVCAR-3) resulted in significantly increased levels of NEAT1, whereas HuR knockdown led to a reduction of NEAT1 [55]. This suggests that the increased levels of NEAT1 found in ovarian cancer could, at least partly, be caused by the elevated levels of the RBP HuR and its stability-promoting effect [55].

A similar scenario was discovered for HuR and lncRNA-HGBC (lncRNA highly expressed in gallbladder carcinoma) by Hu and colleagues [56]. LncRNA-HGBC was first identified in a microarray-based analysis to be highly upregulated in gallbladder cancer tissue from nine patients and was then shown to be linked to gallbladder cancer proliferation and invasion in vitro as well as in vivo [56,83]. Using an RNA pulldown assay, where the in vitro transcribed biotin-labeled lncRNA-HGBC was incubated with lysate from a gallbladder cancer cell line and pulled down with streptavidin beads followed by mass spectrometry analysis of the associated proteins, HuR was identified as an interaction partner of lncRNA-HGBC [56]. This interaction was further verified with an RIP assay [56]. Using different fragments of the lncRNA, the authors were able to pinpoint the binding site to a region containing an ARE [56]. The knockdown of HuR in gallbladder cancer cells led to increased decay of lncRNA-HGBC, suggesting that by shielding the identified ARE of lncRNA-HGBC, HuR stabilizes the cancer-associated lncRNA and thereby potentially contributes to gallbladder carcinogenesis [56].

Contrary to the stabilizing function of HuR that is commonly observed there are also reports that HuR can promote the degradation of certain RNA interaction partners like the ~3-kb-long p53-regulated lncRNA-p21, which is generally downregulated in cancer [29,57]. In an RIP assay carried out with lysate from HeLa cells, Yoon et al. found lncRNA-p21 to be enriched in the HuR immunoprecipitate [57]. Upon further investigating the interaction between lncRNA-p21 and HuR, the authors found that, in contrast to what they expected, small interfering RNA (siRNA)-mediated knockdown of HuR in HeLa cells increased the steady-state level and the half-life of lncRNA-p21 [57]. The mechanism behind this effect was, as the study concluded, that HuR promoted the recruitment of the miRNA let7 and the argonaute-2 (Ago2) protein, a component of the RNA-induced silencing complex (RISC), resulting in the degradation of lncRNA-p21 [57,84]. In line with this, the presence of an antagomir of let7 led to an increase of the levels of lncRNA-p21 despite HuR overexpression, highlighting that the destabilizing effect of HuR on lncRNA-p21 is mediated via let7–Ago2 [57]. The same mechanism was subsequently uncovered for HOTAIR [58]. HOTAIR is transcribed from the antisense strand of the *HOXC* gene cluster and is one of the most studied oncogenic lncRNAs [13,30]. One of many mechanisms of action of HOTAIR is genome-wide transcriptional regulation, for example by recruitment of polycomb repressive complex 2 (PRC2) proteins [13]. By performing an in vitro biotinylated pulldown assay combined with RNA digestion, followed by quantitative reverse transcription PCR (RT-qPCR) analysis of the fragments of HOTAIR that were, due to being bound by HuR, protected from digestion, the binding sites of HuR on HOTAIR were mapped [58]. The results showed that HuR was most frequently bound between positions 1028 and 1272 of the 2158-nt-long lncRNA [13,58]. This sequence is, in general, AU-rich (https://www.ncbi.nlm.nih.gov/gene/; gene ID 100124700), and thus in accordance with the binding specificity of HuR. Why is it then that HuR, in this case, does not contribute to increased stability by its usual mechanism of interfering with the binding of RNA decay-promoting proteins to these AU-rich recognition sites? One explanation could be that, in the case of lncRNA-p21 and HOTAIR, HuR binds to AU-rich sequences that do not conform with the classical ARE motif, which minimally consists of UUAUUUAUU, and which are thus not serving as target sites for decay-promoting RBPs like ARE/poly(U)-binding/degradation factor 1 (AUF1) and tristetraprolin (TTP) [85]. It is also possible that the classical mechanism of HuR to block other RBPs is of no further consequence in cases where the RNA interaction partner additionally contains a binding site for let7, whose recruitment is favored by HuR. HuR–let7–Ago2-mediated RNA degradation via RISC very likely outweighs any RNA stabilizing effects exerted by HuR.

The RBP HuR is an important post-transcriptional regulator that can modulate RNA stability both positively and negatively by either inhibiting or promoting the interaction with other RBPs or miRNAs. What also makes HuR particularly interesting is the fact that its binding sites are very frequent across the genome, as has for example been shown by a transcriptome-wide analysis using photoactivatable ribonucleoside enhanced crosslinking and immunoprecipitation (PAR-CLIP), a method that, first of all, is distinguished from RIP as it includes a UV crosslinking step, thereby reducing the detection of non-specific RNA–protein interactions and that, secondly, incorporates nucleotide analogs which enable a more precise determination of the interaction site between RNA and RBP [82,86]. This PAR-CLIP study identified around 26,000 HuR binding sites, many of which were intronic [86]. Another study by Bakheet et al. found that, in fact, 25% of all human introns contain AREs and that HuR is the RBP that most frequently recognizes these intronic AREs [85]. This high number of HuR binding sites, particularly in intronic regions, suggests that there might be far more lncRNAs than identified so far for which stability is regulated by HuR. Given the frequent overexpression of HuR in many malignancies, therein might also lie a considerable contribution to the deregulation of lncRNAs in cancer.

### 2.2. Serine/Arginine-Rich Splicing Factor 1 (SRSF1)

Serine/arginine-rich splicing factor 1 (SRSF1) is another RBP that is commonly overexpressed in cancer and that exerts oncogenic effects, for example in glioma, by regulating splicing, RNA stability, and nuclear export [60,87]. SRSF1 contains two RRMs at its C-terminal end, a canonical RRM followed by a pseudo-RRM, with the latter playing a more prominent part in determining substrate specificity [88]. The pseudo-RRM binds to RNA in a sequence-specific manner, namely to a GGA motif, that unlike canonical RRMs does not involve the β-sheet surface of the protein but a single α-helix [89]. CLIP analysis combined with high-throughput sequencing has identified a large number of SRFS1 binding sites across different classes of RNA transcripts, including ncRNAs [90].

A study by Zhou et al. aimed to investigate the mechanistic involvement of SRSF1 in gliomagenesis [60]. By RNA-sequencing (RNA-seq) they discovered that the knockdown of SRSF1 in two glioma cell lines resulted not only in differential expression of mRNAs but also of ncRNAs [60]. From a list of 14 lncRNAs that were deregulated in both cell lines, one candidate in particular caught their attention, namely NEAT1, which was downregulated by the SRSF1 knockdown [60]. RIP assays showed that SRSF1 is a direct interaction partner of both NEAT1 isoforms and SRSF1 knockdown was found to result in faster degradation of NEAT1, highlighting a stabilizing role of SRSF1 for NEAT1 [60]. Therefore, the oncogenic lncRNA NEAT1 seems to be independently stabilized by two different RBPs, by HuR and SRSF1, which are both frequently overexpressed in cancer [55,60]. Unlike HuR, the mechanism by which SRSF1 promotes the stability of NEAT1 is not known, but likely follows a similar scheme.

### 2.3. ARE/Poly(U)-Binding/Degradation Factor 1 (AUF1)

The name AUF1 encompasses four different proteins that originate from the *HNRNPD* gene and are generated by alternative splicing [91]. All four AUF1 isoforms form dimers and contain two tandem RRMs that include RNP sequence motifs [91]. Via these domains AUF1 primarily, but not exclusively, binds to U-rich RNA sequences of AREs, therefore competing with HuR [61,71,72,73,85,91]. In addition, it has been found to bind to non-canonical U-rich and GU-rich sites [61,91]. Upon binding, AUF1 promotes the decay of its RNA target [91]. Cytoplasmic mRNA degradation is primarily initiated and also rate-limited by deadenylation, meaning by the excision of the poly(A) tail, which is mostly performed by one of two major deadenylase complexes, carbon catabolite repressor 4–negative on TATA (CCR4–NOT) and chromatin assembly factor 1–negative on TATA (CAF1–NOT) [91]. There is no evidence so far that AUF1 is associated with one of these two complexes, but it is generally believed to facilitate the assembly of other deadenylase complexes [91,92]. The post-transcriptional regulation of mRNA by AUF1 is more complex however, and AUF1 binding does not only induce RNA degradation but can also have quite the contrary effect, namely stabilizing mRNA or promoting translation [93,94]. Accordingly, also the implications of AUF1 in cancer are diverse and it has been found to exert both tumorigenic as well as tumor-suppressive effects [92,95,96,97].

Previously, two examples were given for RBPs that stabilize NEAT1 [55,60]. AUF1, on the other hand, was found to destabilize this tumor-promoting lncRNA [61]. From a PAR-CLIP analysis in HEK293 cells, Yoon et al. identified NEAT1 as a target of AUF1, which was validated by a RIP assay in HeLa cells and an in vitro binding assay [61]. Subsequently, knockdown of AUF1 was observed to increase the half-life of NEAT1 and the number of nuclear paraspeckles, which were distributed more diffusely throughout the nucleus than in control cells, where they clustered into foci [61]. By regulating the stability and the subnuclear localization of NEAT1, AUF1 was also found to affect the nuclear export of some NEAT1 target mRNAs [61].

Apart from NEAT1, Yoon et al. also confirmed MALAT1 as an interaction partner of AUF1 [61]. MALAT1, as mentioned in the introduction, is yet another lncRNA that has been studied intensely in recent years due to its involvement in cancer, for example by regulating alternative splicing [20,30]. AUF1 did not, despite interacting with it, affect the stability of MALAT1 [61]. It is not yet clear why targeting by AUF1 has a destabilizing effect on NEAT1 but no apparent impact on the stability of MALAT1. As the post-transcriptional effects exerted by AUF1 are diverse, it is likely that they are context-specific, meaning that the consequences of AUF1 binding to an RNA target could depend on the RNA sequence, RNA structure, post-transcriptional modifications of AUF1 itself, or the competition/interaction with other RBPs [91]. There are, thus, still many open questions regarding AUF1. Considering the fact that AUF1 binding sites in the genome have been found to be very frequent, with 86,833 being even more numerous than HuR binding sites, and that an astounding 66.8% of these are located in introns, AUF1 is most likely regulating far more lncRNAs than identified so far [61]. Answering the open questions regarding the mechanisms behind AUF1 function and uncovering further lncRNA targets of AUF1 are therefore worthwhile areas for future research.

### 2.4. Polyadenylate-Binding Protein 1 (PABPN1)

PABPN1 is a nuclear RBP that binds to the 3′ poly(A) tail of RNA and, in doing so, stimulates poly(A) synthesis by the poly(A) polymerase [62]. A study lead by Beaulieu et al. focused on investigating the impact of PABPN1 on overall gene expression by conducting an RNA-seq analysis of PABPN1-depleted HeLa cells [62]. Interestingly, they observed that the lack of PABPN1 did not affect the expression level of most mRNAs but had a greater impact on the levels of polyadenylated lncRNAs, with 60 being upregulated more than 2-fold and 16 being downregulated [62]. As the number of upregulated lncRNAs was higher, the study then focused on these candidates [62]. Many of the upregulated lncRNAs were still uncharacterized, but there were also a few more prominent examples, like NEAT1 and TUG1 [62]. TUG1, which stands for taurine upregulated 1, is a lncRNA that has been found to be overexpressed in numerous types of cancer, like small cell lung cancer, cervical cancer, and hepatocellular cancer, and to contribute to various aspects of cancer biology, like proliferation, invasion, metastasis, apoptosis, differentiation, and drug resistance [35,98,99,100]. The transcript that was upregulated the most, and was therefore picked by the authors for further investigations, was the lncRNA small nucleolar RNA, C/D box 60 host gene (SHG60), the gene of which, as the name implies, also encodes, in an intron, the small nucleolar RNA, C/D box 60 (SNORD60) [62]. Importantly, SNORD60 was not upregulated by the PABPN1 knockdown, but only the spliced and polyadenylated SHG60 transcript, indicating that the regulation by PABPN1 occurred on the post-transcriptional level rather than on the transcriptional level [62]. Indeed, PABPN1 was found to promote the degradation of SHG60 in a polyadenylation-dependent manner, in cooperation with the RNA helicase mRNA transport 4 (hMTR4)/superkiller viralicidic activity 2-like 2 (SKIV2L2) [62]. hMTR4/SKIV2L2 is an important coactivator of the RNA exosome, a nuclear RNA degradation machinery consisting of a catalytically inactive nine-subunit core that associates with two ribonucleases, ribosomal RNA processing protein 6 and 44 (hRRP6 and hRRP44) [63,101,102]. Apart from coactivators, the exosome also requires the help of adaptor proteins that guide the entire complex to its RNA targets [63,101,102]. Coactivators and adaptors of the exosome often appear in large complexes, like the nuclear exosome targeting (NEXT) complex or the poly(A) tail exosome targeting (PAXT) connection [63,101]. The PAXT connection has been discovered more recently and contains the zinc finger C3H1-type containing (ZFC3H1) protein, which functions as a linker between PABPN1 and hMTR4 [63]. ZFC3H1 thereby completes the pathway that connects the poly(A)-binding PABPN1 to the nuclear exosome [63]. Via their interaction with different RBPs the NEXT and the PAXT complex guide the exosome towards different types of RNAs [63]. While NEXT, via the RNA-binding motif protein 7 (RBM7), generally targets shorter unprocessed RNAs, PAXT targets longer polyadenylated transcripts, like lncRNAs, via the interaction with PABPN1 [62,63]. The PABPN1–NEXT–exosome pathway thus represents one possibility of how nuclear polyadenylated lncRNAs, among them also cancer-associated lncRNAs like TUG1 and NEAT1, are targeted for degradation [62,63].

In the case of NEAT1, its interaction with PABPN1 is, besides its interaction with AUF1, already the second mechanism that is known to promote its decay. Altogether, NEAT1 is an excellent example of a cancer-associated lncRNA, which is collectively regulated by multiple RBPs that can either enhance its stability (HuR and SRSF1) or promote its degradation (AUF1 and PABPN1). The interplay between all these different RBPs thereby constitutes a very important post-transcriptional mechanism that fine-tunes the level and thus activity of NEAT1.

### 2.5. Insulin-Like Growth Factor 2 mRNA-Binding Protein 1 (IGF2BP1)

IGF2BP1, together with IGF2BP2 and IGF2BP3, belongs to a family of RBPs that is expressed in embryonic tissue and frequently reactivated in different types of cancers but rarely expressed in normal adult tissue [103,104]. Due to its regulation of stability, localization, and translation of numerous mRNAs like c-Myc [105,106], beta-catenin [107], and KRAS [108], it plays a role in embryogenesis and tumor development [103,104]. IGF2BP1 carries two RRMs in its N-terminal region and four hnRNP-K homology (KH) domains in the C-terminus, which are primarily responsible for its interaction with RNA targets in an N6-methyladenosine (m6A)-dependent manner [103,104].

The m6A modification is a reversible and the most abundant RNA modification and stems from the addition of a methyl group to the N-6 position of adenosine by proteins like methyltransferase-like 3 (METTL3), METTL14, and Wilms’ tumor 1-associating protein (WTAP) [109,110]. These proteins form a complex that recognizes and interacts with the consensus motif RRACH (R = G/A and H = A/C/U), a motif that is not only frequently found in mRNA but also in lncRNAs, particularly in circRNAs [109,110]. There are various RBPs, like IGF2BP and the family of YTH domain-containing proteins, that, so to say, read m6A modifications, selectively bind to m6A-modified sites, and subsequently regulate RNA stability, alternative splicing, transport, and translation [109,110]. Hence, the proteins responsible for generating m6A modifications, the ones reading them, like IGF2BP1, and those removing them, namely fat mass and obesity-associated protein (FTO) and AlkB homolog 5 (ALKBH5), altogether impact not only mRNAs but also lncRNAs on the post-transcriptional level [109,110,111]. As this includes oncogenes and tumor suppressors alike, aberrant levels of the m6A modification are linked to tumorigenesis and cancer progression, and have thus received quite some interest in research lately [109,110,111]. For example, in hepatocellular carcinoma (HCC) the methyltransferase METTL3 is elevated and contributes to tumor progression by generating higher levels of m6A modification in the suppressor of cytokine signaling 2 (SOCS2) mRNA which results in its increased degradation mediated by the YTH N6-methyladenosine RNA-binding protein 2 (YTHDF2) [112]. The knowledge on how proteins generating, reading, or erasing m6A modifications regulate the function of specific cancer-related lncRNAs is yet still very limited. MALAT1, for example, has been found to contain an m6A modification site [113]. The methylation at this site induces a structural change in a stem-loop that facilitates binding of the heterogeneous nuclear ribonucleoprotein C (hnRNPC), an RBP that affects mRNA stability, splicing, and export [113]. The effect of hnRNPC binding on MALAT1 has however not been identified so far [113].

IGF2BP1 presents an exception of an m6A-binding RBP with a documented effect on a cancer-associated lncRNA, namely the ~500-nt-long lncRNA highly up-regulated in liver cancer (HULC) [64]. HULC, as described by its name, is “highly up-regulated in liver cancer” and contributes to carcinogenesis, not only in the liver, by acting as a ceRNA [26,64,114,115]. While the expression of HULC in HCC was previously described to be induced by the transcription factor cAMP response element-binding protein (CREB), there was not much known about the regulation of HULC on the post-transcriptional level [64]. Consequently, Hämmerle and colleagues aimed to investigate the role that RBPs play in regulating this lncRNA [64]. For this purpose, they employed an RNA affinity purification assay in which they identified the IGF2BP family as specific interaction partners of HULC [64]. However, only IGF2BP1 was found to have an impact on the stability of the lncRNA [64]. Knockdown of the RBP in HepG2 cells almost doubled the half-life of HULC, whereas overexpression decreased its level, indicating a destabilizing effect of IGF2BP1 on the HCC-associated lncRNA [64]. Upon trying to understand the mechanism behind this effect, the authors of the study found that IGF2BP1 interacted with CCR4-NOT transcription complex subunit 1 (CNOT1), a component of the deadenylase complex CCR4-NOT, which induces 3′-5′-decay of cytoplasmic RNAs [64]. Interestingly, IGF2BP1 had previously only been known to increase the stability of its RNA targets, as exemplified by c-Myc, which is shielded from the degradation trough endoribonucleases by being sequestered in so-called messenger ribonucleoproteins (mRNPs) that are formed by IGF2BP1 together with various other RBPs [103,104,105,106]. The study by Hämmerle et al. was the first to report a destabilizing function exerted by IGF2BP1 [64]. Why does IGF2BP1 stabilize most of its mRNA targets but promote the degradation of the lncRNA HULC? Whether it shields an RNA or subjects it to degradation via the CCR4–NOT complex is assumed to depend on the contribution of other protein interaction partners, like CNOT1 or the mRNP components, whose involvement, in turn, might rely, on the one hand, on sequence or structural motifs present in the RNA target and, on the other hand, on post-translational modifications of IGF2BP1 itself [104]. For example, it was found that Scr-mediated phosphorylation of IGF2BP1 could induce the release of an mRNA from mRNPs [116]. That post-translational modifications of IGF2BP1 might play a crucial role in regulating the activity of this RBP was also suggested by the findings of Hämmerle and colleagues [64]. They did not observe a negative correlation between the mRNA level of IGF2BP1, which is in fact upregulated rather than downregulated in HCC, and the level of HULC, which is highly elevated in HCC but whose decay was found to be promoted by IGF2BP1 [64,103]. This indicates that the activity of IGF2BP1 might somehow be impeded in HCC by post-translational mechanisms [64,103]. Given the importance of IGF2BP1 in cancer development, for example by regulating c-Myc and HULC, future studies will most likely put a stronger focus on how this RBP is regulated. Moreover, as the m6A consensus sequence, to which IGF2BP1 binds, is a rather frequent motif also in lncRNAs, further cancer-associated lncRNAs that interact with IGF2BP1 might be identified in the future. Also, the impact of other m6A-binding RBPs on cancer-related lncRNAs will likely receive more focus in future research, especially in light of the recent development of an improved method that allows quantitative profiling of m6A modifications at single-nucleotide resolution [117].

### 2.6. Tristetraprolin (TTP)

Besides the two previously discussed examples HuR and AUF1, TTP is another RBP that binds to AREs, adenylate/uridylate-rich RNA motifs that function as signals for the rapid degradation of RNA [61,66,71,72,73,85,91]. While HuR promotes the stability of RNAs by shielding these recognition sites, AUF1 recruits, upon binding to AREs, components of the RNA degradation machinery [71,72,73,91]. The exact components are not yet known [91]. In case of TTP, on the other hand, it is well documented that it interacts with CCR4–NOT, one of the two major deadenylase complexes responsible for RNA decay, as well as with PM/Scl-75, a component of the exosome [66]. In addition, TTP also interacts with PABPN1, thereby interfering with polyadenylation, and is involved in mRNA decapping [66]. In summary, TTP destabilizes its RNA targets via multiple different pathways [66]. As there are numerous oncogenes amongst the targets of TTP, it is generally regarded as a tumor suppressor [66]. Accordingly, TTP is frequently downregulated in cancer and its low expression or inactivation by phosphorylation is associated with cancer development and progression as well as with poor patient prognosis [66]. Examples for oncogenes that are destabilized by TTP at the mRNA level are Twist1 and Snail1 [118], Cyclin D1 and c-Myc [119], and Bcl-2 [120]. There is also one cancer-associated lncRNA that has been reported to be destabilized by TTP, namely HOTAIR [67]. Even though this relation was not reported in the context of cancer but observed in a study investigating the role of TTP and HOTAIR in trophoblasts [67], it can be assumed that this also holds true for cancerous cells where HOTAIR is generally upregulated and TTP downregulated [13,66]. Consequently, the reduced levels of functional TTP in cancer, and thus reduced RNA degradation, could be a general mechanism contributing to the increased levels of cancer-associated lncRNAs. Future studies will certainly identify more lncRNA targets of TTP.

## 3. RBPs Regulating lncRNA Transport and Localization

### 3.1. Human Antigen R (HuR)

For a lncRNA to fulfill its function in a cell it is not only important how much of this lncRNA is present in a cell but equally, or potentially even more important, whether the lncRNA is properly transported to and located at its site of action. While lncRNAs are frequently enriched in the nucleus, where they are for example involved in transcriptional regulation, they can also be located in the cytoplasm or in mitochondria and can be shuttled between these different compartments in response to different cellular conditions [8,59,121,122]. There is, however, not a lot known about how this transport of lncRNAs is regulated and facilitated [59,121]. The RPB HuR constitutes one exception as it is known to not only regulate the level of lncRNAs by affecting their stability but has also been found to have an impact on lncRNA localization [55,56,57,58,68,77]. The different regulatory mechanisms that HuR exerts on its lncRNA targets are illustrated in Figure 1. 

A study by Noh et al. found HuR to bind to both the 3′ and 5′ end of the RNA component of mitochondrial RNA processing endoribonuclease (RMRP), a 265-nt-long lncRNA that is best known as a component of the mitochondrial RNA-processing endoribonuclease (RNase MRP) complex and that is also involved in mitochondrial DNA replication [68,121]. Rather recently, an additional role of RMRP in different types of cancer has been reported by multiple publications that describe a contribution of RMRP to, for example, cancer cell proliferation, migration, and invasion, by sponging up different miRNAs [123,124,125,126]. According to Noh and colleagues, HuR binds to RMRP already in the nucleus and subsequently facilitates its nuclear export [68]. HuR itself can shuttle back and forth between the nucleus and cytoplasm in response to different stimuli [71,72]. The export of HuR from the nucleus occurs in a chromosomal maintenance 1 (CRM1)-dependent manner [68,127]. CRM1 is a member of the importin β superfamily that facilitates the nuclear export of proteins, to which it can bind either directly or via adaptors, as well as of RNA, in which case adaptor proteins are necessary [59,128]. Noh et al. showed that not only silencing of HuR itself in HEK293 cells but also of CRM1 led to significantly reduced cytoplasmic levels of nascent RMRP [68]. Importantly, there was no additive effect when both HuR and CRM1 were knocked down, highlighting that the nuclear export of RMRP was facilitated via the HuR–CRM1 axis [68]. While affecting the localization of RMRP, HuR was not observed to have an impact on the steady-state levels, meaning the stability, of the lncRNA [68]. As discussed in the first half of the review, there are a number of cancer-related lncRNAs whose stability is regulated by HuR [55,56,57,58,77] and based on the frequency of HuR binding sites it can be anticipated that more will be identified in the future. However, so far, RMRP is the only lncRNA whose nuclear export has been reported to be mediated by HuR.

### 3.2. G-Rich RNA Sequence-Binding Factor 1 (GRSF1)

Even though the lncRNA RMRP is encoded in the nucleus, it is also present in mitochondria [68,121]. As detailed above, Noh et al. uncovered that the first step in the transport pathway of RMRP, the export from the nucleus into the cytoplasm, is facilitated by the RBP HuR [68]. In the same study the authors also reported that the subsequent import of RMRP into mitochondria seemed to be mediated via two import machineries, the TOM/TIM machinery and polynucleotide phosphorylase (PNPase) [68]. The TOM/TIM machinery consists of a translocase complex of the outer mitochondrial membrane (TOM) and two different inner membrane translocases (TIM) that together enable the transport of proteins across both mitochondrial membranes into the matrix [129]. The PNPase is located in the mitochondrial intermembrane space and promotes the import of RNA from the cytoplasm into the mitochondrial matrix [68,130]. How exactly RMRP is transported by these machineries and which RBPs it interacts with in order to do so has not been elucidated [68]. What Noh et al. did report was that the RBP GRSF1, while not directly contributing to its import, promoted the accumulation of RMRP in the mitochondrial matrix [68]. GRSF1 contains three RRMs via which it binds to a G-rich recognition motif (AGGGGD, with D = A/U), subsequently regulating splicing, polyadenylation, and export of its RNA targets [68,121,131]. One isoform of GRSF1 is located in mitochondria where it forms granules with newly synthesized mitochondrial RNAs, among them the two mitochondrial lncRNAs lncCyt b and lncND5, which carry 10 and 21 GRSF1 consensus-binding sites, respectively [121,131]. Knockdown of GRSF1 in immortalized primary fibroblasts decreased the overall level of these two mitochondrial lncRNAs by about half [131]. Noh et al., however, reported that the whole-cell level of RMRP was not affected by GRSF1 knockdown, but only its accumulation in the mitochondrial matrix [68]. Hence, GRSF1 seems to retain RMRP in the matrix once it has been imported, thereby facilitating its enrichment at its specific site of action [68].

RMRP is not the only example of a nuclear-encoded lncRNA that can be transported into mitochondria. MALAT1 was also found to be present in mitochondria [122]. Under normal conditions, MALAT1 is mostly located in the nucleus where it regulates alternative splicing and gene expression [20,122,132]. In HCC, however, Zhao et al. observed an increased level of MALAT1 within mitochondria [122]. An RNA-fluorescent in situ hybridization (FISH) assay combined with mitochondrial staining showed that MALAT1 was highly enriched in HepG2 mitochondria but barely detectable in non-cancerous hepatic cells [122]. This mitochondrial accumulation of the lncRNA in HCC appears to enhance mitochondrial energy metabolism, thus potentially contributing to the oncogenic effects that MALAT1 exerts in HCC [122,133]. Interestingly, the study observed that for the mitochondrial lncRNA lncCyt b the situation is the other way around, namely that in non-cancerous cells it is located primarily in the mitochondrial matrix, where it is bound by GRSF1, but in HepG2 cells, it shows increased presence in the nucleus [122]. How the shuttling of these two lncRNAs between the nucleus and mitochondria is mediated is not known so far. It is clear, however, that different RBPs, like HuR and GRSF1, must play a central role in this process and that deregulations in this network of RBPs are responsible for the aberrant localization of lncRNAs in cancer, as observed for MALAT1 and lncCyt b in HCC. In case of lncCyt b, a disturbed interaction with GRSF1 could be one factor contributing to this phenomenon.

### 3.3. Insulin-Like Growth Factor 2 mRNA-Binding Protein 1 (IGF2BP1)

The oncofetal RBP IGF2BP1 has already been discussed in regard to its destabilizing effect on the HCC-associated lncRNA HULC, which is mediated by the recruitment of the deadenylase complex CCR4–NOT via CNOT1 [64]. In addition to regulating the stability of both mRNA and lncRNA, IGF2BP1 also plays a role in coordinating RNA localization [65,103,104,105,106,107,108,134,135,136,137]. IGF2BP1 is mostly cytoplasmic where it forms, together with its RNA targets and other RBPs, mRNP granules [103,104,105,106,134]. When not associated with these mRNPs, IGF2BP1 can also translocate to the nucleus where it has been found to bind to mRNA already during transcription [135,136]. As IGF2BP1 contains two nuclear export signals (NES) within its second and fourth RNA-binding KH domain it is subsequently exported back into the cytoplasm together with its bound RNA target [135,136]. Hence, IGF2BP1 facilitates the nuclear export of its RNA target [135,136]. The prime example of this is given by beta-actin. IGF2BP1 binds to the beta-actin mRNA as soon as it is transcribed and subsequently enables its export into the cytoplasm, where IGF2BP2 either locates to perinuclear regions or interacts with and moves along the cytoskeleton towards the cell periphery, more precisely towards newly forming lamellipodia [134,136,137]. Here, IGF2BP2 is phosphorylated by the Src-kinase, which results in the disassociation of the beta-actin mRNA and its localized translation [137].

Following the same mechanism as observed for the beta-actin mRNA, IGF2BP1 also regulates the subcytoplasmic distribution of the lncRNA H19 [65]. H19 is a 2.3-kb-long, spliced and polyadenylated lncRNA that is, similarly to IGF2BP1, expressed during embryonic development and reactivated in several types of cancer [65,103,104,138]. H19 shows a diverse range of actions: it functions as a ceRNA, induces or represses the transcription of various genes, and interacts with and thus modulates the activity of different proteins like p53, thereby contributing to all hallmarks of cancer [138,139]. Runge et al. discovered that IGF2BP1 binds to the 3′ end of H19 with high affinity and thereby targets the lncRNA to lamellipodia and perinuclear regions of proliferating mouse embryonic fibroblasts [65]. In growth-arrested confluent cells, the IGF2BP1–H19 complex was dispersed more evenly in the cytoplasm [65]. It has been discovered that H19 seems to contribute to the migratory behavior and branching morphogenesis of epithelial cells, processes that are essential during embryogenic development but that also enable migration, invasion and metastasis of cancerous cells [65,140]. This function of H19 coincides well with the targeting of IGF2BP1-H19 to the leading edge of proliferative cells.

### 3.4. Heterogeneous Nuclear Ribonucleoprotein K (hnRNPK)

The above discussed examples of RBPs are involved in the transport or localization of cancer-associated lncRNAs outside of the nucleus. In general, lncRNAs, however, tend to be enriched in nuclear fractions [8]. A well-studied RBP that plays a role in the nuclear accumulation of lncRNAs is the heterogeneous nuclear ribonucleoprotein K (hnRNPK) [69]. The exact mechanism of how hnRNPK can retain RNAs in the nucleus is however still unknown [69]. HnRNPK fulfills a wide variety of cellular functions, for example by acting as a transcription factor, regulating translation, and serving as a hub for various signaling pathways [141]. Numerous studies have observed oncogenic effects as well as a prognostic relevance of hnRNPK in different types of cancer, like breast, colorectal, and gastric cancer, where its overall levels are increased and it is aberrantly localized in the cytoplasm [142,143,144]. The binding of hnRNPK to RNA occurs via the interaction of its three KH domains, which are a type of RNA-binding motif first discovered in and therefore named after the RBP, with poly-C sequences in the RNA targets [145,146]. Lubelsky and Ulitsky, by screening a library of short fragments from nuclear mRNAs and lncRNAs, discovered a specific consensus sequence that is bound by hnRNPK and mediates nuclear accumulation [69]. This sequence consists of a 42-nt-long fragment that overlaps with Alu repeats, a very common type of short interspersed element (SINE), in antisense orientation and that contains three stretches of at least six pyrimidines (C/T), with two of these stretches matching the consensus sequence RCCTCCC (R = A/G) [69]. They termed the sequence SINE-derived nuclear RNA LOcalizatIoN (SIRLOIN) [69].

SIRLOINs are found in 13.1% of human lncRNAs and 7.5% of mRNAs and contribute substantially to the nuclear enrichment of RNA transcripts [69]. An example of a cancer-associated lncRNA that contains a SIRLOIN and is retained in the nucleus by hnRNPK is MALAT1 [69]. As stated before, the nuclear accumulation of MALAT1 has, however, been found to be disturbed in HCC, where undefined reasons lead to an increased mitochondrial presence of the lncRNA [122]. A recent study by Nguyen et al. showed that the deletion of a SINE in MALAT1 and the hence disrupted interaction with hnRNPK resulted in a more frequent translocation to the cytoplasm [70]. This was accompanied by increased DNA damage and apoptosis due to the redistribution of a protein called transactive response DNA binding protein 43 kDa (TDP-43) to the cytoplasm along with MALAT1 to which it is bound [70]. Aberrant expression and localization of hnRNPK, commonly observed in cancer, could thus also play a role in the change of MALAT1 localization in HCC. As reviewed elsewhere, there are numerous examples for interactions between lncRNAs and hnRNPK where they jointly regulate gene expression by various means [145]. Thus, the deregulation of hnRNPK in cancer has far-reaching ramifications as it affects the function and localization of lncRNAs [145].

## 4. Conclusions

LncRNAs are versatile regulators of basic cellular processes, like transcription, splicing, and translation. There are numerous examples of lncRNAs that are deregulated in various types of cancer, thereby contributing to cancer initiation and progression. In order to better understand these implications of lncRNAs in cancer, it is necessary to understand how lncRNAs are regulated. An important form of regulation of lncRNAs occurs at the post-transcriptional level and is mediated by a network of RNA-binding proteins. In the case of cancer-associated lncRNAs, their regulation by RBPs is of particular interest because RBPs themselves are frequently deregulated in various malignancies and could thus constitute a major contribution to the deregulation of lncRNAs. Despite the existence of a vast number of both RBPs and lncRNAs, so far, only a rather small number of RBPs have been documented to explicitly regulate cancer-related lncRNAs. In this review we chose to discuss several of the interactions between well-known RBPs that also have been found to play a role in cancer, and extensively studied cancer-associated lncRNAs. These interactions, in particular, have an impact on the stability and the transport or subcellular localization of lncRNAs, both of which are essential for a lncRNA to properly fulfill its functions. Post-transcriptional modifications, like m6A, are also playing a role in this regard as they affect the binding of certain RBPs. The mechanisms by which RBPs regulate lncRNAs are diverse and the same RBP can have different effects on different lncRNAs, highlighting their context specificity that, in large parts, depends on the synergy or competition with other RBPs or also miRNAs. A prime example for an RBP that itself is commonly upregulated in cancer and that has been found to regulate multiple cancer-associated lncRNAs by various means is HuR. While stabilizing some, it promotes the degradation of others and has also been implicated in the nuclear export of a lncRNA. As the number of predicted HuR binding sites in the genome, especially in intronic regions, is high, many more lncRNAs than identified so far might be regulated by this RBP. In general, there is still a lot to uncover regarding the regulation of lncRNAs by RBPs. Not much is known yet about how m6A modifications affect the function of specific cancer-associated lncRNAs, how lncRNAs are shuttled between or retained within different cellular compartments, how post-translational modifications of RBPs affect their regulation of lncRNAs, and how different RBPs either compete or collaborate in order to regulate the destiny and function of a target lncRNA.

## Figures and Tables

**Figure 1 ijms-21-02969-f001:**
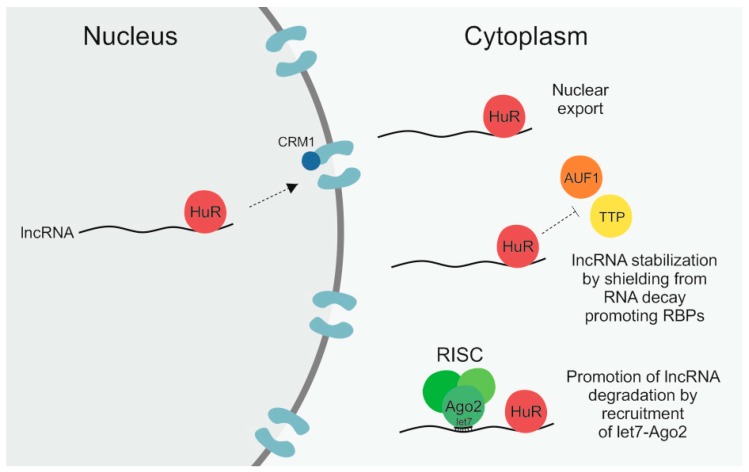
The RNA-binding protein (RBP) human antigen R (HuR) can regulate long non-coding RNAs (lncRNAs) by different means. First, by binding to its target lncRNA in the nucleus it can facilitate its subsequent nuclear export by interaction with the importin β superfamily member chromosomal maintenance 1 (CRM1). Secondly, by shielding lncRNAs from RNA decay promoting RBPs like ARE/poly(U)-binding/degradation factor 1 (AUF1) and tristetraprolin (TTP), HuR can enhance lncRNA stability. Thirdly, by recruitment of let7–Argonaute-2 (Ago2) it can promote lncRNA degradation via the RNA-induced silencing complex (RISC).

**Table 1 ijms-21-02969-t001:** RNA-binding proteins (RBPs) and how they regulate known cancer-associated long non-coding RNAs (lncRNAs). Ago2: argonaute-2; CCR4-NOT: carbon catabolite repressor 4–negative on TATA; CNOT1: CCR4–NOT transcription complex subunit 1; CRM1: chromosomal maintenance 1; HOTAIR: HOX antisense intergenic RNA; HULC: highly up-regulated in liver cancer; lncRNA-HGBC: lncRNA highly expressed in gallbladder carcinoma; MALAT1: metastasis-associated lung adenocarcinoma transcript 1; NEAT1: nuclear-enriched abundant transcript 1; NEXT: nuclear exosome targeting; RMRP: RNA component of mitochondrial RNA processing endoribonuclease; TUG1: taurine upregulated 1.

RNA-Binding Protein	Regulated lncRNA	Regulatory Mechanism	Reference
Human antigen R (HuR)	NEAT1	Stabilization by shielding from RNA decay-promoting proteins	[55]
	lncRNA-HGBC	Stabilization by shielding from RNA decay-promoting proteins	[56]
	lncRNA-p21	Promotion of degradation by recruitment of let7–Ago2	[57]
	HOTAIR	Promotion of degradation by recruitment of let7–Ago2	[58]
	RMRP	Facilitation of nuclear export via interaction with CRM1	[59]
Serine/arginine-rich splicing factor 1 (SRSF1)	NEAT1	Stabilization by an unknown mechanism	[60]
Arginine/uridine-rich RNA element (ARE)/poly(U)-binding/degradation factor 1 (AUF1)	NEAT1	Destabilization, probably by recruitment of a deadenylase complex	[61]
Polyadenylate-binding protein 1 (PABPN1)	NEAT1	Promotion of degradation by recruitment of NEXT–exosome	[62,63]
	TUG1	Promotion of degradation by recruitment of NEXT–exosome	[62,63]
Insulin-like growth factor 2 mRNA-binding protein 1 (IGF2BP1)	HULC	Promotion of degradation by recruitment of the deadenylase complex CCR4–NOT via CNOT1	[64]
	H19	Targeting of lamellipodia and perinuclear regions	[65]
Tristetraprolin (TTP)	HOTAIR	Promotion of degradation, most likely by recruitment of the deadenylase complex CCR4–NOT	[66,67]
G-rich RNA sequence-binding factor 1 (GRSF1)	RMRP	Retention in the mitochondrial matrix by an unknown mechanism	[68]
Heterogeneous nuclear ribonucleoprotein K (hnRNPK)	MALAT1	Retention in the nucleus by an unknown mechanism	[69,70]

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
