# Peer review of "RNA-Binding Proteins as Important Regulators of Long Non-Coding RNAs in Cancer"

_ijms, 2020, doi:10.3390/ijms21082969_

Round 1

Reviewer 1 Report

The review manuscript entitled: "RNA-Binding Proteins as Important Regulators of Long Non-Coding RNAs in Cancer" presents a timely overview on a relevant topic, namely the impact of RBPs on lncRNAs. It is generally well-written and well-structured and will make a fine contribution to the IJMS special issues after some minor issues haven been addressed.

1) The authors should briefly describe how they selected the reviewed examples - given that there are more than 1600 lncRNAs linked to cancer in lnc2cancer [30] and certainly more than 1000 RBPs in the human proteome, there are certainly more (potential) regulatory interactions. Hence, it should be explicitly stated how the examples here were selected and there should be a brief statement in the conclusions that this review not necessarily comprehensively covering all regulatory RBP-lncRNA interactions.

Other examples could also be added, e.g. METTL16 binding the stabilizing triple helix structure of MALAT1 (PMID 27872311).

2) For the central lncRNAs covered in the review, the original discovery papers should be cited, not only the database [30] in line 73.

3) The gene name for AUF1 is HNRNPD, which should be mentioned somewhere to allow its recognition by search algorithms.

4) For each discussed RBP, the authors should at a brief sentence what is known about the binding specificity (e.g. for sequence or structure recognition) for this RBP, to allow an estimation whether the described regulatory RBP-lncRNA interaction is likely specific or likely applicable to a larger number of lncRNAs.

5) For the lncRNA localization, multiple recent studies have identified nuclear RNA localization signals also in lncRNAs as well as RBPs associated with them, which should be included - e.g. HNRNPK (PMID 29466324) - or the ribosome association of cytoplasmic lncRNAs (PMID 27090285).

6) The authors only discuss the impact of RBPs on lncRNA stability or lncRNA transport / localization. What about lncRNA modifications? These should be added in a separate chapter.

Author Response

1) The authors should briefly describe how they selected the reviewed examples - given that there are more than 1600 lncRNAs linked to cancer in lnc2cancer [30] and certainly more than 1000 RBPs in the human proteome, there are certainly more (potential) regulatory interactions. Hence, it should be explicitly stated how the examples here were selected and there should be a brief statement in the conclusions that this review not necessarily comprehensively covering all regulatory RBP-lncRNA interactions.

Other examples could also be added, e.g. METTL16 binding the stabilizing triple helix structure of MALAT1 (PMID 27872311).

Response: In the introduction we made the following change to address this point (starting from line 83):

The RBPs that are discussed in this review (for a summary see Table 1) were thus chosen because they are, just like the lncRNAs they regulate, known to play a role in cancer. Even though these examples do not cover all regulatory interactions between RBPs and lncRNAs they include well-known RBPs and extensively studied cancer-associated lncRNAs and give a comprehensive overview of the two main aspects of post-transcriptional regulation of lncRNAs, namely lncRNA stability on the one hand and lncRNA transport and localization on the other hand.

In the conclusion we made the following amendments (starting from line 519), also to refer to the issue addressed under point 6:

Despite the existence of a vast number of both RBPs and lncRNAs, so far, only a rather small number of RBPs have been documented to explicitly regulate cancer-related lncRNAs. In this review we chose to discuss several of the interactions between well-known RBPs, that also have been found to play a role in cancer, and extensively studied cancer-associated lncRNAs. These interactions, in particular, have an impact on the stability and the transport or subcellular localization of lncRNAs, both of which are essential for a lncRNA to properly fulfill its functions. Also post-transcriptional modifications, like m6A, and the RBPs binding to them affect lncRNA function.

2) For the central lncRNAs covered in the review, the original discovery papers should be cited, not only the database [30] in line 73.

Response: Where the lncRNAs were mentioned first, we have added the original discovery papers as references (see line 72, 73): According to lnc2Cancer, the most studied cancer-associated lncRNAs are MALAT1 [46], H19 [47], HOTAIR [48], MEG3 [49], TUG1 [50], ANRIL [51] and NEAT1 [52].

3) The gene name for AUF1 is HNRNPD, which should be mentioned somewhere to allow its recognition by search algorithms.

Response: We have added the gene name of AUF1 in line 200 (The name AUF1 encompasses four different proteins that originate from the HNRNPD gene and…).

4) For each discussed RBP, the authors should at a brief sentence what is known about the binding specificity (e.g. for sequence or structure recognition) for this RBP, to allow an estimation whether the described regulatory RBP-lncRNA interaction is likely specific or likely applicable to a larger number of lncRNAs.

Response: The following information regarding the binding specificity of each RBP was either already included in the review previously or adapted/added now:

HuR (already included previously):

Starting in line 91: The human antigen R (HuR), the protein product of the ELAV1 gene, is a ubiquitously expressed RBP that contains three RNA recognition motives (RRMs) via which it preferentially binds to adenylate/uridylate-rich RNA elements (AREs).

Starting in line 162: What also makes HuR particularly interesting is the fact that its binding sites are very frequent across the genome as has, for example, been shown by a transcriptome-wide analysis using PAR-CLIP (photoactivatable ribonucleoside enhanced crosslinking and immunoprecipitation), … This PAR-CLIP study identified around 26,000 HuR binding sites, many of which are intronic [74]. Another study by Bakheet et al. found that, in fact, 25 % of all human introns contain AREs and that HuR is the RBP that most frequently recognizes these intronic AREs [73]. This high number of HuR binding sites, particularly in intronic regions, suggests that there might be far more lncRNAs than identified so far, whose stability is regulated by HuR. Given the frequent overexpression of HuR in many malignancies, therein might also lie a considerable contribution to the deregulation of lncRNAs in cancer.

SRSF1 (added now, starting line in 179):

SRSF1 contains two RRMs at its C-terminal end, a canonical RRM followed by a pseudo-RRM, with the second one playing a more prominent part in determining substrate specificity [77]. The pseudo-RRM binds to RNA in a sequence-specific manner, namely to a GGA motif, that unlike canonical RRMs does not involve the β-sheet surface of the protein but a single α-helix [78]. CLIP analysis combined with high-throughput sequencing has identified a large number of SRFS1 binding sites across different classes of RNA transcripts, including ncRNAs [79].

AUF1 (already included previously):

Starting in line 201: All four AUF1 isoforms form dimers and contain two tandem RRMs that include RNP sequence motifs [80]. Via these domains AUF1 primarily, but not exclusively, binds to U-rich RNA sequences of AREs, therefore competing with HuR [55–57,73,80,81]. In addition, it has been found to bind to non-canonical U-rich and GU-rich sites [80,81].

Starting in line 231: Considering the fact that AUF1 binding sites in the genome have been found to be very frequent, with 86,833 being even more numerous than HuR binding sites, and that an astounding 66.8 % of these are located in introns, AUF1 is most likely regulating far more lncRNAs than identified so far [81].

PABPN1 (already included previously, starting in line 237):

PABPN1 is a nuclear RBP that binds to the 3` poly(A) tail of RNA and, in doing so, stimulates poly(A) synthesis by the poly(A) polymerase [88].

IGF2BP1 (already included previously, slightly adapted now):

Starting in line 281: IGF2BP1 carries two RRMs in its N-terminal region and four hnRNP-K homology (KH) domains in the C-terminus which are primarily responsible for its interaction with RNA targets in an N6-methyladenosine (m6A)-dependent manner [95,96]. The m6A modification is a reversible and the most abundant RNA modification and stems from the addition of a methyl group to the N-6 position of adenosine by proteins like methyltransferase-like 3 (METTL3), METTL14, and Wilms’ tumor 1-associating protein (WTAP) [101,102]. These proteins form a complex that recognizes and interacts with the consensus motif RRACH (R = G/A and H = A/C/U), a motif that is not only frequently found in mRNA but also in lncRNAs, particularly in circRNAs [101,102].

Starting in line 338: Moreover, as the m6A consensus sequence, to which IGF2BP1 binds, is a rather frequent motif also in lncRNAs, further cancer-associated lncRNAs that are interacting with IGF2BP1 might be identified in the future

TTP (already included previously, starting in line 345):

Besides the two previously discussed examples HuR and AUF1, TTP is another RBP that binds to AREs, adenylate/uridylate-rich RNA motifs that function as signals for the rapid degradation of RNA [55–57,73,80,81,111].

GRSF1 (already included previously, starting in line 412)

GRSF1 contains three RRMs via which it binds to a G-rich recognition motif (AGGGGD, with D = A/U), subsequently regulating splicing, polyadenylation and export of its RNA targets [116,119,128].

hnRNPK (added now, starting in line 488):

The binding of hnRNPK to RNA occurs via the interaction of its three KH domains, a type of RNA-binding motif first discovered in and therefore named after the RBP, with poly-C sequences in the RNA targets [144,145]. Lubelsky and Ulitsky, by screening a library of short fragments from nuclear mRNAs and lncRNAs, discovered a specific consensus sequence that is bound by hnRNPK and mediates nuclear accumulation [139]. This sequence consists of a 42-nt long fragment that overlaps with Alu repeats, a very common type of short interspersed element (SINE), in antisense orientation and that contains three stretches of at least six pyrimidines (C/T), with two of these stretches matching the consensus sequence RCCTCCC (R = A/G) [139]. They termed the sequence SIRLOIN (SINE-derived nuclear RNA LOcalizatIoN) [139]. SIRLOINs are found in 13.1% of human lncRNAs and 7.5% of mRNAs and contribute substantially to the nuclear enrichment of RNA transcripts [139].

5) For the lncRNA localization, multiple recent studies have identified nuclear RNA localization signals also in lncRNAs as well as RBPs associated with them, which should be included - e.g. HNRNPK (PMID 29466324) - or the ribosome association of cytoplasmic lncRNAs (PMID 27090285).

Response: We have now included a chapter on hnRNPK and the SIRLOIN motif, an important consensus sequence in lncRNAs that is bound by hnRNPK and associated with their nuclear localization. We intended to include hnRNPK previously because it is, just like the other RBPs that were chosen to be discussed in the review, a well-studied RBP with implications in cancer.

6) The authors only discuss the impact of RBPs on lncRNA stability or lncRNA transport / localization. What about lncRNA modifications? These should be added in a separate chapter.

Response: Instead of adding a separate chapter we have adapted/extended the section on the m6A modification under the chapter 2.5 of IGF2BP1 (see below), a RBP known to bind to this modification. The m6A modification is the most abundant modification in both mRNA and lncRNA, and to our knowledge most studied and most relevant modification in lncRNAs. Nevertheless, the number of studies giving examples of RBPs generating, reading or erasing m6A modifications that, in doing so, specifically regulate cancer-associated lncRNAs are still very limited. In addition, as seen in the example of IGF2BP1, these RBPs, in the end, may also regulate the stability or localization of lncRNAs. We therefore did not think it necessary to include a separate chapter on RBPs that generate or bind to lncRNA modifications, simply because we believe there are not enough examples and because they cannot be clearly separated from other RBPs that regulate the stability or localization of lncRNAs.

Starting in line 284: The m6A modification is a reversible and the most abundant RNA modification and stems from the addition of a methyl group to the N-6 position of adenosine by proteins like methyltransferase-like 3 (METTL3), METTL14, and Wilms’ tumor 1-associating protein (WTAP) [101,102]. These proteins form a complex that recognizes and interacts with the consensus motif RRACH (R = G/A and H = A/C/U), a motif that is not only frequently found in mRNA but also in lncRNAs, particularly in circRNAs [101,102]. There are various RBPs, like IGF2BP and the family of YTH domain-containing proteins, that, so to say, read m6A modifications, selectively bind to m6A-modified sites and subsequently regulate RNA stability, alternative splicing, transport and translation [101,102]. Hence, the proteins responsible for generating m6A modifications, the ones reading them, like IGF2BP1, and those removing them, namely fat mass and obesity-associated protein (FTO) and AlkB homolog 5 (ALKBH5), altogether impact not only mRNAs but also lncRNAs on the post-transcriptional level [101–103]. As this includes oncogenes and tumor suppressors alike, aberrant levels of the m6A modification are linked to tumorigenesis and cancer progression, thus receiving quite some interest in research lately [101–103]. For example, in hepatocellular carcinoma (HCC) the methyltransferase METTL3 is elevated and contributes to tumor progression by generating higher levels of m6A modification in the SOCS2 mRNA which results in its increased degradation mediated by the m6A-binding protein YTHDF2 [104]. The knowledge on how proteins generating, reading or erasing m6A modifications regulate the function of specific cancer-related lncRNAs is yet still limited. MALAT1, for example, has been found to contain an m6A modification site [105]. The methylation at this site induces a structural change that facilitates binding of the heterogeneous nuclear ribonucleoprotein C (hnRNPC), a RPBP that affects mRNA stability, splicing and export [105]. The effect of hnRNPC binding on MALAT1 has however not been identified so far [105].

Starting in line 340: Also, the impact of other m6A binding RBPs on cancer-related lncRNAs will likely receive more focus in future research, especially in light of the recent development of an improved method that allows profiling of m6A modifications at single-nucleotide resolution [110].

In the conclusion, starting in line 534: In general, there is still a lot to uncover regarding the regulation of lncRNAs by RBPs. Not much is known yet about how m6A modifications affect the function of specific cancer-associated lncRNAs, how lncRNAs are shuttled between or retained within different cellular compartments, how post-translational modifications of RBPs affect their regulation of lncRNAs and how different RBPs either compete or collaborate in order to regulate the destiny and function of a target lncRNA.

Reviewer 2 Report

The review is overall well written and organized. It comprehensively covers a relatively new topic (how RBPs regulate long noncoding RNAs), so I think it can be a useful tool for readers approaching the field. In light of this, it would be useful to add a table/insert/figure that summarizes and briefly explains the methods used to measure interaction between proteins and RNAs (RIP, CLIP, pulldown).

Some minor corrections are needed:

- ARE are cited in the text as arginine uridine rich motifs, but it should be adenylate/uridylate rich;

- references are numbered twice in the reference list.

Author Response

1. The review is overall well written and organized. It comprehensively covers a relatively new topic (how RBPs regulate long noncoding RNAs), so I think it can be a useful tool for readers approaching the field. In light of this, it would be useful to add a table/insert/figure that summarizes and briefly explains the methods used to measure interaction between proteins and RNAs (RIP, CLIP, pulldown).

Response: We have added a brief explanation of each of the methods where they were mentioned first. In addition, we refer the readers to a review that comprehensively covers these and other methods currently used to study the interaction between proteins and lncRNAs.

  • Starting in line 109: RIP assays present a simple and widely used method to study RNA-protein interactions (for a comprehensive review on this and other methods to study RNA-protein interactions see [70]) and are based on the use of antibodies to precipitate and isolate a protein of interest together with its associated RNAs [70].
  • Starting in line 120: Using an RNA pulldown assay, where the in vitro transcribed biotin-labeled lncRNA-HGBC was incubated with lysate from a gallbladder cancer cell line and pulled down with streptavidin beads, followed by mass spectrometry analysis of the associated proteins, HuR was identified as an interaction partner of lncRNA-HGBC [62].
  • Starting in line 164: …PAR-CLIP (photoactivatable ribonucleoside enhanced crosslinking and immunoprecipitation), a method that, first of all, discerns itself from RIP as it includes a UV crosslinking step, thereby reducing the detection of non-specific RNA-protein interactions and that, secondly, incorporates nucleotide analogs which enable a more precise determination of the interaction site between RNA and RBP [70,74].

2. ARE are cited in the text as arginine uridine rich motifs, but it should be adenylate/uridylate rich;

Response: Arginine/uridine-rich RNA element has been changed to adenylate/uridylate-rich both in line 93 and line 346.

3. References are numbered twice in the reference list

Response: This was a problem that apparently arose during the process of editing and has now been corrected. Thank you for drawing our attention to it!